# ON THE IMPORTANCE OF DISTRACTION-ROBUST REPRESENTATIONS FOR ROBOT LEARNING

## ABSTRACT

Representation Learning methods can allow the application of Reinforcement Learning algorithms when a high dimensionality in a robot's perceptions would otherwise prove prohibitive. Consequently, unsupervised Representation Learning components often feature in robot control algorithms that assume high-dimensional camera images as the principal source of information. In their design and performance, these algorithms often benefit from the controlled nature of the simulation or the laboratory conditions they are evaluated in. However, these settings fail to acknowledge the stochasticity of most real-world environments. In this work, we introduce the concept of Distraction-Robust Representation Learning. We argue that environment noise and other distractions require learned representations to encode the robot's expected perceptions rather than the observed ones. Our experimental evaluations demonstrate that representations learned with a traditional dimensionality reduction algorithm are strongly susceptible to distractions in a robot's environment. We propose an Encoder-Decoder architecture that produces representations that allow the learning outcomes of robot control tasks to remain unaffected by these distractions.

## 1 INTRODUCTION

Representation Learning techniques form an integral part in many Reinforcement Learning (RL) robot control applications (Lesort et al., 2018). Utilising low-dimensional representations can allow for a faster and more efficient learning of tasks than when using high-dimensional sensor information (Munk et al., 2016). This is particularly useful in vision-based learning when high-dimensional images of the robot's environment are the principal source of information available to the learning algorithm (Zhu et al., 2020). Most commonly, representations are learned by applying dimensionality reduction techniques such as Autoencoders (AEs) (Hinton & Salakhutdinov, 2006) or Variational Autoencoders (VAEs) (Kingma & Welling, 2014) to the robot's sensory data (Lange et al., 2012; Zhu et al., 2020). Generally, an AE consists of two Neural Networks, an Encoder $E$ and a Decoder $D$. The Encoder attempts to condense all available information in the input data $\boldsymbol{x}$ into a latent representation $\boldsymbol{z}$ from which a reconstruction of the inputs $D(E(\boldsymbol{x}))$ is generated by the Decoder. When the dimensionality of the representation is smaller than that of the input data, some information is lost when creating the representations. An AE is typically trained to shrink the magnitude of this information loss by minimising a reconstruction error. This error is commonly given by the squared norm of differences,

$$L_{\text{AE}} = ||\boldsymbol{x} - D(E(\boldsymbol{x}))||_2^2. \tag{1}$$

However, the optimisation of the reconstruction error in Eq. 1 does not necessarily result in the generation of representations that are optimal for the use in robot learning algorithms. For example, an accurate reconstruction of the decorative patterns on a dinner plate is less important than the plate's dimensions to a robot learning to place it into a cabinet. It can therefore be desirable to control which aspects of the information contained in the inputs are most critical to be preserved in the representations. For instance, Pathak et al. (2017) design a Neural Network that learns representations from visual inputs by using them to predict the action taken by the RL agent in its state transition. By asking the network to predict the action, the authors eliminate the requirement for the representations to retain any state information that is unrelated to the agent's behaviour.

A focus on the learned representations' preservation of task-relevant information becomes even more crucial in the presence of *distracting influences* (DIs) in the environment. These DIs can materialise in the presence of additional environment objects which exhibit dynamics that are not only uncorrelated with the robot's behaviour but additionally misleading. For instance, a robot that is learning to move objects to different positions in a room can find the observation of a moving autonomous vacuum cleaner misleading. Alternatively, DIs can impact the dynamics of existing objects in the room. For instance, after the robot has moved a box to a certain position in the room, further movements of the box due to external forces can be distracting to the robot's learning process.

In this paper, we introduce the concept of *Distraction-Robust Representation Learning*. We investigate the learning outcomes of robot control tasks when DIs are present in the environment. We show that in the presence of DIs, representations learned exclusively from environment observations can mislead the robot's perceptions of its control over the environment. This finding demonstrates that Distraction-Robust Representation Learning needs to be afforded increased attention. In particular, works in the strand of research that aim to make RL algorithms more applicable to real-world scenarios largely concentrate on improving algorithm attributes such as the data efficiency (Zhu et al., 2020). However, few works acknowledge the challenges posed by the inherently stochastic nature of real-world environments and the presence of DIs (Forestier et al., 2017). Furthermore, we introduce a Robot Action Encoder-Decoder architecture (RAED) which successfully produces representations that are robust to DIs in the environment. RAED follows the simple but effective approach of using only the values that parameterise the robot's actions as the input to the Encoder. Such a set of parameters defines a robot controller for instance. The representations produced by the Encoder are used by RAED's Decoder to generate predictions of the environment observations. RAED's design allows for static environment elements to be learned by the Decoder while concentrating the information in the representations on the observable consequences of the robot's behaviour. Moreover, when environment observations are distorted by the presence of DIs, RAED produces representations that capture the expected consequences of the robot's environment interactions. This is not the case when training representation learning methods such as AEs to reconstruct the full content of the robot's visual perceptions. We can therefore draw some parallels between RAED's design and the concept of a forward model (Jordan & Rumelhart, 1992). Given the simplicity of the approach, we expect the applicability of RAED to generalise to various different learning algorithms.

## 2 RELATED WORK

Several works have investigated mechanisms to preserve only task-relevant information in learned representations. Pathak et al. (2017) propose a Neural Network architecture that learns representations of visual inputs by predicting the action taken by the RL agent in its state transition. This design allows the representations to dedicate their information capacity to the observable consequences of the robot's actions. Finn et al. (2016) propose a spatial AE to learn representations that aim to preserve only the configuration of objects in the environment rather than all aspects of the information contained in the camera images. However, in both approaches, the representations are learned from visual inputs which will be distorted if DIs are present in the environment. Without an explicit correction mechanism, these representation learning techniques therefore remain susceptible to distractions. The concept of affordance learning formulates a similar goal in discovering the consequences of the robot's actions on its environment (Cakmak et al., 2007; Şahin et al., 2007). However, the works in this strand of research rarely consider the problem of DIs in the environment. Instead, they mainly concentrate on the robot's ability to infer how an object in the environment would behave in response to its actions when no prior interaction experience with that particular object is available (Dehban et al., 2016; Mar et al., 2015).

A work that uses learned representations and investigates robot interactions in the presence of DIs is presented in the Intrinsically Motivated Goal Exploration Processes (IMGEP, Laversanne-Finot et al. (2018)). IMGEP aims to enable robots to explore the possible interactions with various tools in an environment that also features distractor objects. These objects either cannot be interacted with or move independently of the robot. The authors show that a variant of their proposed algorithm remains unaffected by the presence of these distractors. This robustness is demonstrated by the robot's lack of interaction with the distractor objects. However, the DIs we consider in this paper pose an arguably larger challenge for two main reasons. First, we evaluate distractor objects which exhibit dynamics that are not only independent of the robot's behaviour but also misleading to the robot's

perceptions of its interactions with the actual objects of interest. For instance, a distractor object can collide with one of the objects manipulated by the robot. Second, we consider the case of misleading external DIs on the interesting objects themselves. In this case, the object interaction cannot be avoided. Furthermore, in IMGEP the distractor robustness is not learned by the VAE that generates the representations. Instead, the algorithm identifies the components in the representations that correspond to the distractors during the robot's interaction phase. Indeed, the learned representations remain static in this phase as the VAE is pre-trained on a manually generated dataset that covers all possible environment observations by spawning the various environment objects, including the distractors, in different positions.

Alternative exploration-based approaches have been proposed recently (Forestier et al., 2017; Pathak et al., 2017; Eysenbach et al., 2019; Sharma et al., 2019). A particularly interesting framework is presented in AURORA (Cully, 2019), which we use in our experimental evaluations. Instead of decoupling the representation learning phase from the interaction phase as done in IMGEP, AURORA proposes a joint approach to discover a diverse set of behaviours of any robot in any given environment without the need to specify a task objective or reward function. More precisely, AURORA applies a dimensionality reduction algorithm such as an AE to the robot's sensory data that are collected during the execution of the discovered controllers. The learned latent representation associated to a controller's collected sensory data defines the Behavioural Descriptor (BD) for this controller. By measuring the distance between different controllers' BDs, the novelty of a given behaviour can be determined. The BD can therefore be used to build a collection of controllers that each exhibit different behaviours. AURORA uses this collection to further explore the space of possible behaviours via stochastic mutations, and continuously adds additional novel controllers to the collection. Periodically, the AE is trained using the sensory data that are extracted from all controllers in the collection.

The AURORA framework is well-suited to judge the impact of DIs on the learned representations. Indeed, AURORA's decision to accept newly generated controllers into the collection is exclusively based on the distance between their BDs. We can therefore analyse the influence of DIs using metrics that are formulated directly in the latent representation space. These metrics are more sensitive to the impact of DIs than measures of the overall learning outcomes. This is because the learning outcomes are typically also influenced by components of the learning algorithm other than the representations. Furthermore, AURORA's training of the dimensionality reduction technique provides a particularly challenging setting for RAED. The training dataset composition is continually evolving as the data is collected online. Additionally, AURORA rejects controllers that generate behaviours very similar to that of existing controllers. This complexifies the task of learning the expected consequences of the robot's actions when the collected environment observations are distorted by DIs.

## 3 Method

In this work, we use a single vector $\boldsymbol{a}$ to parameterise the set of actions taken by the robot from the beginning until the end of the simulation. This vector of the *robot's actions* can store the parameter values of a controller for instance. We use a vector $\boldsymbol{x}$ to parameterise the single environment observation that is returned by each simulation. This vector can store the pixel values of an image for example.

RAED consists of an Encoder-Decoder structure that uses $\boldsymbol{a}$ and $\boldsymbol{x}$ to produce a representation $\boldsymbol{z}$ that is distraction-robust. More precisely, the latent representation $\boldsymbol{z}$ is generated from RAED's Encoder network $E$ which takes the robot's actions $\boldsymbol{a}$ as its inputs. This conditioning of the representations on the robot's actions is the crucial element in RAED's design. The Decoder uses the representations to produce a prediction of the environment observation $D(E(\boldsymbol{a}))$. The actual environment observation $\boldsymbol{x}$ is used as the prediction target in the training of RAED. This yields the prediction error given in Eq. 2. Note, the difference in the Encoder input compared to an AE's reconstruction error in Eq. 1.

$$C = ||\boldsymbol{x} - D(E(\boldsymbol{a}))||_2^2. \tag{2}$$

Depending on the specifics of the robot design and the task, a set of controllers with different parameters can yield the same observed behaviour. For instance, when steering a robot arm with redundancies in its joints, different controllers can achieve the same position in a gripper connected

to the arm. For such a set of controllers, it is possible that the differences in their parameters lead to the generation of different representations when using RAED, even if the observation predictions are exactly the same. This is less likely to occur in a dimensionality reduction technique that takes the visual perceptions as its inputs. For these methods, the principal sources of variation in the representations are the differences in the observations rather than the differences in the robot's actions. To reduce RAED's freedom to disperse the representations of similar observed behaviours, we investigate the addition of a penalty term $P$. The overall training loss is then given by $L_{\mathrm{RAED}} = C + P$.

One possible instantiation of $P$ is the Kullback-Leibler divergence (KL) term used in a VAE (Kingma & Welling, 2014). This KL term implicitly imposes a penalty on the magnitude of the means produced by the VAE's Encoder network. The consequence is a crowding of the representations into a smaller latent subspace. This can have the desired effect of reducing RAED's freedom to produce dissimilar representations for similar observed behaviours. However, the practical implementation of the KL term also imposes a penalty on any Encoder variance magnitudes different than 1. Large Encoder variances result in an increased dispersion of representations when these are sampled from the Encoder. The presence of these variances can therefore be interpreted as a further source of stochasticity when learning representations. Our experimental results in Section 4.4 demonstrate that the application of the KL term benefits from leaving the Encoder variance outputs unused. The derived penalty can be summarised in a simple linear term given by $P_{\mathrm{linear}} = ||z||_2^2$.

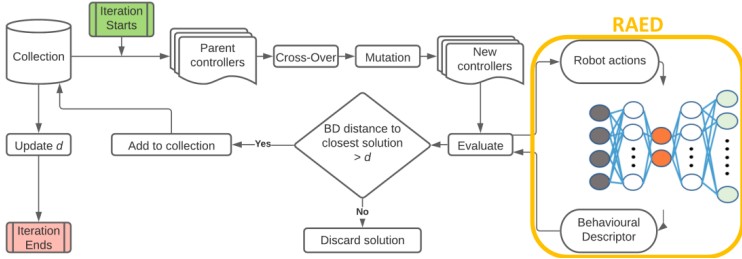

Figure 1: AURORA algorithm with a modular integration of RAED.

In the AURORA framework, RAED can be used as a modular representation learning component to produce the BDs for the discovered controllers. We present an illustration of AURORA in Fig. 1 and a pseudo-code in Algorithm 1 in Appendix B.1. AURORA largely follows a standard Quality-Diversity algorithm (Cully & Demiris, 2018a). At the start of each iteration, a predefined number of controllers is selected from the collection. This group of controllers forms the set of parent controllers. Each pair of parents undergoes cross-over and mutation operations. Cross-over operations produce new controllers that bear similarities to both parents. Mutation operations introduce some stochastic changes to these new controllers. If the collection is empty at the start of the iteration, we instead generate new controllers by randomly sampling in the controller space.

Each controller is executed in the simulated environment and the associated observations are captured. These observations are used to produce the BD of each controller. When using an AE in AURORA, each BD is the latent representation generated using the environment observation $x$ as input. In RAED, the robot's actions $a$ are the inputs instead. AURORA, therefore, stores for each controller a vector of robot action parameters, an environment observation and a BD.

In the decision to accept a given new controller into the collection, AURORA considers the distance between its BD to the descriptor of the nearest controller contained in the collection. If this distance exceeds a minimum distance threshold $d$, the controller is deemed sufficiently novel to be added to the collection (Cully & Demiris, 2018b). Otherwise, it is discarded. $d$ is initialised at the start of the algorithm using the BDs of the initial set of controllers contained in the collection. AURORA then updates the value of $d$ after each iteration to steer the size of the collection towards the target size.

Every $N$ iterations, AURORA extracts the environment observations and robot actions from all controllers contained in the collection to train the AE or RAED respectively. Once the training of the network concludes, all controllers are removed from the collection and assigned new BDs. This set of controllers is finally re-added to the collection using the acceptance mechanism described in the previous paragraph. This concludes one iteration of the AURORA algorithm. The hyperparameters used in AURORA are given in Appendix B.1.

# 4 EXPERIMENTS

In our experimental evaluations, we study the impact of DIs on RAED and on an AE that uses visual perceptions as its inputs. The learned representations are two-dimensional and used by AURORA to generate a diverse set of robot behaviours. We consider an experiment setup of a planar robot arm with four degrees of freedom mounted in the center of a table. The arm interacts with environment objects in two different scenarios: 1) an object arrangement scenario, and 2) an air-hockey scenario. Each scenario evaluates a different type of DI which we detail in the following paragraphs. Fig. 2 presents the simulated environment and an illustrative environment observation for each scenario. The configuration and hyperparameters of the simulations are given in Appendix C.

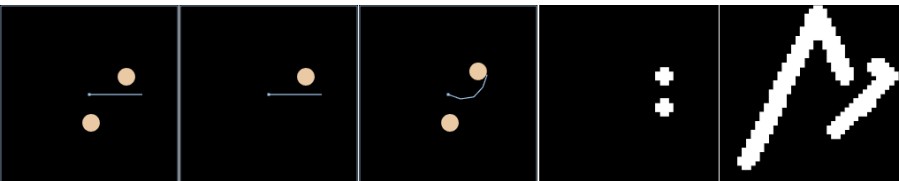

Figure 2: Environment simulations and observations. From left to right: (a) Object arrangement scenario initial simulation state without any distracting influences (DIs). (b) Air-hockey scenario initial simulation state without DIs. (c) Robot arm pushing an object in the environment. (d) Object arrangement environment observation with DIs. (e) Air-hockey environment observation with DIs.

## 4.1 SCENARIO 1: OBJECT ARRANGEMENT

The object arrangement scenario features two objects positioned on the table as shown in Fig. 2a. The robot arm can push these objects as shown in Fig. 2c but they do not move easily due to their large weight. Each controller is two-staged and consists of 8 parameter values. The first 4 parameters dictate the angles in the 4 joints of the arm to be reached by the end of the first half of the simulation. The latter 4 dictate the angles to be reached by the end of the second half. The robot's environment interactions are simulated for 10 seconds. The environment observation consists of a 40 x 40 image that captures the objects' positions at the end of the simulation. The robot arm itself does not feature in the observations. The DIs materialise in an external force that pushes one of the two objects in a random direction at the start of the simulation. This DI simulates the presence of a meddling human while our household robot is learning to arrange objects in the environment. We simulate several DI frequency levels which we refer to as the *environment stochasticity*. At 0.5 environment stochasticity, there is a 50% probability that the external force is applied in any given simulation. The object to be pushed is selected randomly. Fig. 2d presents an environment observation after a DI has shifted the lower object from its starting position.

## 4.2 SCENARIO 2: AIR-HOCKEY

This scenario features an air-hockey puck that spawns in the same starting location at the beginning of each simulation as shown in Fig. 2b. The robot arm can strike the puck as illustrated in Fig. 2c. The puck glides on the table surface and can bounce off the walls. A controller now consists only of 4 parameters which dictate the angles in the joints of the arm to be reached by the end of the simulation. The environment observation consists of a 40 x 40 long-exposure image that traces out the trajectory of the puck. The robot arm does not feature in the observations. DIs are simulated by the addition of a second puck. This puck appears in a random position at the start of the simulation and moves in a random direction at a random velocity. This "random" puck is also captured in the environment observation. At 0.5 environment stochasticity, the random puck appears in any given simulation with 50% probability. This DI simulates the presence of an additional player with their own puck while the robot is perfecting its air-hockey game. Fig. 2e shows an environment observation of the random puck's trajectory in the bottom right of the image and the non-random puck's trajectory to its left.

### 4.3 PERFORMANCE METRICS

We calculate two performance metrics: 1) the behavioural diversity, and 2) the proportion of *no-move controllers*. We perform 5 repetitions of each evaluation to obtain robustness in the results.

The behavioural diversity is a common way to evaluate the quality of AURORA's learning outcome (Cully, 2019). It measures the diversity of the behaviours that are generated by all controllers in the collection. We quantify this diversity as the variance in the positions of the objects of interest. In the arrangement scenario, we extract both environment objects' positions at the end of the simulation. We calculate the variance in each object's position across all controllers in the collection and average these two scores to obtain our metric. In the air-hockey scenario, we extract for each controller the non-random puck's trajectory coordinates. For each trajectory element, we calculate the average variances in the x and y-coordinates across all controllers in the collection to obtain a score. The overall diversity in the behaviours is measured as the average of the scores over all trajectory elements. In both scenarios, a larger variance score therefore shows an improved learning outcome. If DIs affect the learned representations, controllers can be misdescribed by their BDs. Very similar controllers can then be accepted repeatedly into the collection. A susceptibility to DIs would therefore show in decreasing variance scores as the environment stochasticity increases.

In both scenarios, some arm movements do not produce any change in the objects' positions. Since the robot arm itself is not captured in the environment observation, such "no-move" controllers produce the same observation when no DIs are present in the environment. Their BDs are therefore identical and AURORA allows at most one such no-move controller to be present in the collection at any given time. However, controllers that do not move any objects in a distraction-free environment do not appear as such whenever DIs are present. If the DIs distort these controllers' BDs, AURORA considers them different from other no-move controllers. Consequently, a presence of DIs can fool AURORA into accepting multiple no-move controllers into the collection. We therefore use the proportion of no-move controllers in the collection as a second metric to quantify the impact of the DIs on the learned representations. If this metric increases in the environment stochasticity, we conclude that the generation of representations is susceptible to DIs in the environment.

### 4.4 EXPERIMENTAL RESULTS

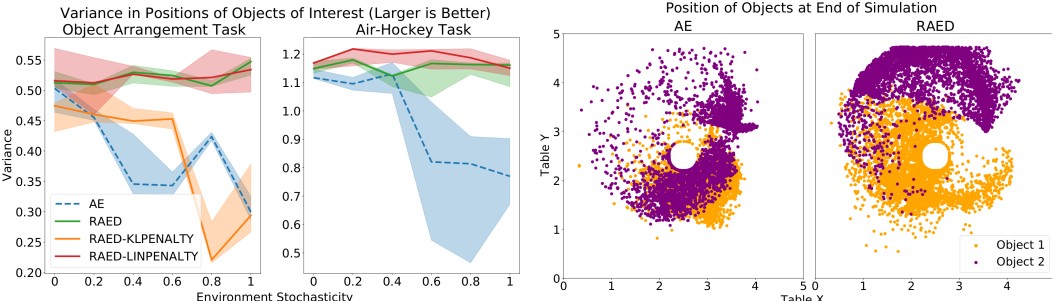

Figure 3: The variance in the environment objects' positions. Left (a): Object arrangement task. Right: (b) Air-hockey task. The lines represent the median scores while the shaded areas extend to the first and third quartile of the data distribution obtained over 5 repetitions.

Figure 4: Positions of the two objects in the arrangement task after the execution of all controllers in two collections generated at 100% environment stochasticity. AURORA achieves a larger exploration in the object configurations when using RAED.

Fig. 3 shows the behavioural diversity achieved in both scenarios when using the AE, RAED and two additional variants of RAED in which we add the KL term and the linear penalty term $P_{\text{linear}}$ from Section 3 respectively. Note, that the AE and RAED's variants have a larger network capacity in the air-hockey scenario than in the arrangement task. This is to account for the more complex observation type of a long-exposure image in the air-hockey task, compared to the snapshot of the environment objects' positions in the arrangement scenario (see Fig. 2). In both scenarios, the use of RAED's learned representations leads to stable variance scores in Figs. 3a and 3b across all

evaluated DI frequency levels. This stability demonstrates that RAED's representations are robust to distractions and encode the expected consequences of the robot's actions rather than the observed ones. Conversely, the use of an AE leads to decreasing trends in the variance scores in Figs. 3a and 3b. These negative trends indicate that the robot's perceptions of its behaviour are distorted in the AE's learned representations due to the DIs in the environment. As a result of this distortion, controllers that produce very similar object configurations in the arrangement task, and very similar puck trajectories in the hockey task, are assigned BDs that make them appear different from each other. Consequently, when using the AE, AURORA repeatedly accepts similar controllers into the collection. This susceptibility to distractions leads to decreasing variance scores for the AE as DIs grow more frequent. At 100% environment stochasticity, the use of an AE leads to median variance scores of 0.31 and 0.78 in the arrangement and air-hockey task respectively. These are significantly lower than the scores of 0.55 and 1.08 when using RAED. In both scenarios, the two methods' differences in the learning outcomes can be seen in visualisations of the exploration in the achievable behaviours. At 100% environment stochasticity, the use of RAED leads to a wider exploration in the possible object configurations in the arrangement task in Fig. 4. Fig. 5 shows a substantially larger spread in the trajectories of the non-random air-hockey puck when using RAED. Similarly, large differences can be observed between the AE and RAED methods in their network outputs. We provide some illustrative examples in Appendix A.

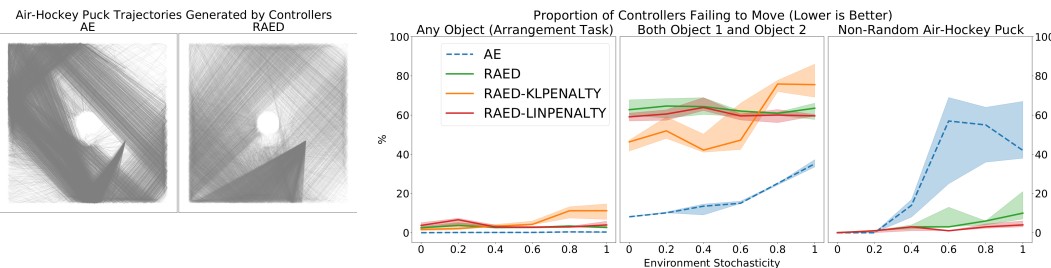

Figure 5: Puck trajectories generated by controllers in the collection at 100% environment stochasticity. RAED achieves a larger spread in the trajectories.

Figure 6: From left to right: Proportion of controllers failing to move (a) any object of interest in the object arrangement task, (b) both Object 1 and 2 in the object arrangement task, (c) the non-random puck in the air-hockey task. The lines and shaded areas are obtained as described in Fig. 3.

Fig. 6 shows the proportion of controllers in the collection that fail to move the objects of interest in the two scenarios. As explained previously, an increasing trend in this metric indicates the learned representations' sensitivity to the presence of DIs. In the arrangement task, RAED produces a stable score across all environment stochasticities in Fig. 6a. This affirms the distraction-robustness of RAED's learned representations. In the air-hockey task, there is only one object that the robot needs to learn to control. Compared to the arrangement scenario, this increases the proportion of all controllers that lead to no movement in the object of interest. This increases the complexity of the task and leads to a slightly positive trend in Fig. 6c when using RAED. Nevertheless, RAED's learned representations are sufficiently distraction-robust for AURORA to maintain a stable diversity in the generated collections as we saw in Fig. 3b. As expected, the AE's scores are more strongly affected by the DIs in the environment. In the air-hockey task, Fig. 6c shows a rapidly increasing proportion of no-move controllers. Similarly, Fig. 6b shows a rising proportion of controllers failing to move both objects in the arrangement task. However, the proportion of controllers failing to move any object in this task when using the AE provides an interesting result. Fig. 6a shows that at 0% environment stochasticity, the use of the AE's learned representations leads to a minimal proportion of no-move controllers. This is expected, as such a distraction-free environment does not provide any challenges to the AE. Surprisingly, however, even as the environment stochasticity grows larger, the use of the AE's representations persistently leads to low proportions of no-move controllers. This stability appears to indicate that the learned representations are insusceptible to the DIs in the environment. However, the AE variant's sharply decreasing variance scores in Fig. 3a show a strong sensitivity to the environment distractions in this task. The reconciliation of these two contradicting observations can be found in the setup of the arrangement scenario. In this task, the likelihood for any given controller to produce a movement in at least one of the objects is very large for two reasons. First, both objects are placed in close proximity to the robot arm. This can

be seen in the depiction of the initial simulation state in Fig. 2a. Second, the robot executes a two-staged controller. Consequently, even when DIs distort the perception of the robot's behaviour, most discovered controllers will still produce a movement in at least one of the objects. This explains the AE variant's minimal proportion of no-move controllers across environment stochasticities.

Fig. 6c further shows that at 0% environment stochasticity the use of RAED in the air-hockey task results in the same minimal proportion of no-move controllers as the use of the AE. However, RAED's score in the arrangement task is slightly larger in Fig. 6a. In this scenario, the AE and RAED variants produce scores of 0.01 and 2.53 percentage points respectively. This disparity is explained by the larger number of parameters in the controllers used in the arrangement task. In a distraction-free environment, all no-move controllers generate the same observed behaviour. However, each controller has different parameters which serve as the inputs to RAED's Encoder. These differences can therefore allow RAED to produce a different BD for each of these no-move controllers. In this case, AURORA would believe these controllers to generate different behaviours. As a result, a larger proportion of no-move controllers is accepted into the collection. In the air-hockey task, the controllers only have 4 parameter values. The variability in RAED's inputs is therefore much more limited than in the arrangement task, where controllers consist of 8 parameter values which makes a dispersion in the BDs more likely. However, one can note that RAED's marginally larger proportion of no-move controllers in the arrangement scenario does not affect the overall learning outcome of AURORA. The AE and RAED achieve the same variance scores at 0% environment stochasticity in Fig. 3a.

One approach to reducing RAED's freedom to disperse the representations lies in the use of a penalty term. In RAED-KLPENALTY, we evaluate a VAE that takes the robot's actions as its inputs. The Encoder network now produces the mean and variance of a normal distribution. As discussed in Section 3, we can interpret the KL divergence of this produced distribution and a standard Gaussian distribution as a penalty term on the size of the latent space. However, the results in Figs. 3a and 6a show that the application of this penalty in the object arrangement scenario results in a rapid deterioration in both metrics as the environment stochasticity increases. This is explained by the KL term's pressure on the network to increase Encoder variances up to a magnitude of 1. Large variance values lead to a greater dispersion in the learned representations when these are sampled from the Encoder. The Encoder variance term can therefore be interpreted as a further source of stochasticity in the generation of the learned representations. We therefore omit the RAED-KLPENALTY variant from our evaluation in the air-hockey task. Instead, we create the RAED-LINPENALTY configuration by removing the Encoder variances. This simplifies the KL term to the penalty term $P_{\text{linear}}$ that we introduced in Section 3. In the air-hockey task, the introduction of this linear penalty term leads to a lower proportion of no-move controllers in Fig. 6c and a larger variance score in Fig. 3c. In the arrangement scenario, on the other hand, the use of RAED and RAED-LINPENALTY produces approximately the same scores in both performance metrics in Figs. 3a and 6a. This suggests that the efficacy of this penalty term depends on the specifics of the task and the observation type. We can introduce a weight to control the effective cost of the penalty in order to adapt it to the task at hand. However, any result shown here on the optimality of a given magnitude in this weight would be specific to our experiments. We therefore omit any further evaluations of different weight values, as we want to avoid any such task-specificity in this work in order to preserve RAED's more generally applicable concept of using the robot's actions to learn a representation.

## 5 CONCLUSION

In this paper, we introduced the concept of Distraction-Robust Representation Learning. We demonstrated the necessity of this approach in two experimental scenarios. The use of an Autoencoder led to a large distortion in the robot's perceived behaviour when distracting influences were present in the environment. The susceptibility to environment distractions led to poor learning outcomes when using the Autoencoder's learned representations. We introduced RAED, an Encoder-Decoder architecture which learns representations that allowed the generation of controller collections to become robust to the distracting influences considered in both scenarios. Finally, we also show that the number of controllers that do not interact with the objects in the environment can be reduced with the introduction of a linear penalty term in the loss function.

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

## A  NETWORK PREDICTIONS

We can inspect the networks' predictions to obtain a qualitative assessment of the learned representations' distraction-robustness. In the following visualisations, we show the networks' predictions and the corresponding environment observations in the air-hockey task at 100% environment stochasticity. In this task, the presence of DIs materialises in an additional object in the environment. This allows a clear distinction of the observation elements that are present due to the DIs. In the object arrangement task, the DIs are more difficult to distinguish in the observations. We therefore do not show any visualisations from that experiment scenario here.

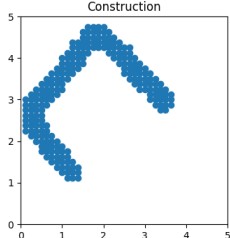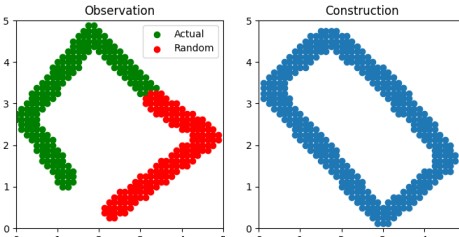 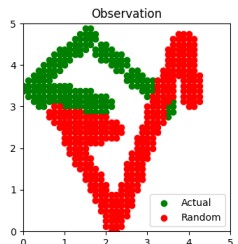

Figure 7: Air-hockey scenario at 100% environment stochasticity. Left: Prediction made by RAED. Right: Corresponding environment observation showing a particularly misleading environment observation. The distinction between the "actual" puck and the random puck is made for analysis purposes and not available to the network. Pixel activations range from 0 to 1 and are thresholded at a value of 0.5 for the illustration.

Figure 8: Air-hockey scenario at 100% environment stochasticity. Left: Prediction made by RAED. Right: Corresponding environment observation showing a collision between the two pucks. Pixel activations are thresholded as in Fig. 7.

Fig. 7 shows the environment observation and associated prediction made by RAED. The distinction between the actual non-random puck and the random puck in the observation is made only for analysis purposes and is not available to the network. All pixels take values in the range of [0, 1]. The prediction visualisation is thresholded at a value of 0.5. The environment observation shows that the two pucks' trajectories connect to form a longer trajectory. This type of physically plausible observation is one of the most difficult types of observations for RAED to predict. However, we can see that RAED's prediction accurately traces out the non-random puck's trajectory, while ignoring the random puck entirely. Furthermore, by encoding the expected perceived consequences of the robot's actions, RAED is able to impact of collisions between the two pucks. Fig. 8 shows a collision between the two pucks at the table coordinates (1, 3). Nevertheless, RAED's prediction accurately traces out the expected trajectory instead.

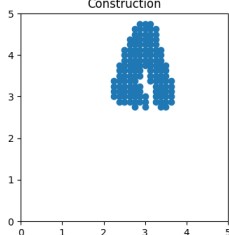 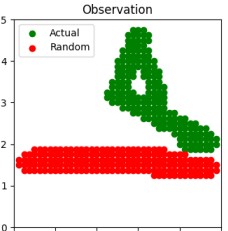

Figure 9: Air-hockey scenario at 100% environment stochasticity. Left: Prediction made by RAED. Right: Corresponding environment observation showing the non-random puck colliding with the robot arm. Pixel activations are thresholded as in Fig. 7.

Another challenging situation for RAED is posed by the puck colliding with the robot arm. Small changes in the controller can lead to large differences in the shape of the arm and therefore the trajectory of the puck after impact. For these cases, we observe in the worst case that RAED's un-

certainty results in a truncated trajectory prediction. We show an example in Fig. 9. Note, however, that RAED successfully excludes the trajectory of the non-random puck from the prediction.

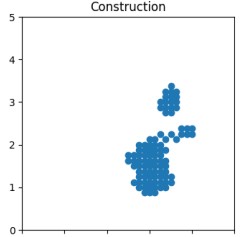 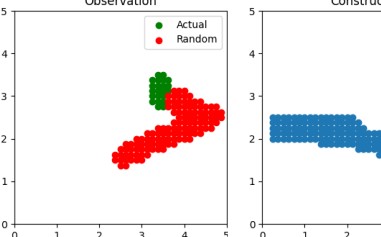 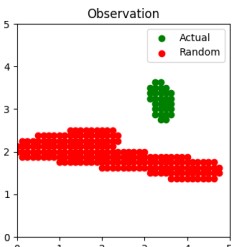

Figure 10: Air-hockey scenario at 100% environment stochasticity. Left: Prediction made by AE. Right: Corresponding environment observation showing two unconnected puck trajectories. Pixel activations are thresholded as in Fig. 7.

Figure 11: Air-hockey scenario at 100% environment stochasticity. Left: Prediction made by AE. Right: Corresponding environment observation showing two unconnected puck trajectories. Pixel activations are thresholded as in Fig. 7.

The AE method's predictions are strongly affected by the presence of DIs. Every prediction we inspected, features elements of the non-random puck's trajectory. In the best case, the two trajectories are not connected and the non-random puck trajectory is not reconstructed with full certainty. Fig. 10 shows an example of this. This is due to the limited variability in the achievable trajectories of the non-random puck. Certain areas of the table are frequented largely or even exclusively by the random puck only. The low frequency of observations that feature activated pixels in these areas, allows the AE to ignore the reconstruction of some of these observation elements. However, in the worst case, the non-random puck transverses areas of the table that are frequently activated in the environment observations. Fig. 11 shows that in these situations, the AE accurately reconstructs the full trajectory of the non-random puck. This demonstrates the strong susceptibility of the AE's learned representations to the DIs in the environment.

## B   IMPLEMENTATION DETAILS

### B.1   AURORA

We provide a pseudo-code of AURORA in Algorithm 1. Tab. 1 presents the hyperparameters used in the configuration of AURORA. Our AURORA implementation is built using the Sferes2 framework (Mouret & Doncieux, 2010).

---

**Algorithm 1** Pseudocode of the AURORA algorithm (Cully, 2019).

---

1: **procedure** AURORA($num\_iterations, training\_frequency, collection\_target\_size$)
2:     $collection \leftarrow random\_controllers$   ▷ Initialise collection with set of random controllers
3:     $L \leftarrow initialise\_L(collection)$                       ▷ Initialise distance threshold L
4:     **for** iteration in num_iterations **do**
5:         $parents \leftarrow select(collection)$              ▷ Select set of parent controllers
6:         $new\_controllers \leftarrow mutate(cross\_over(parents))$
7:         $generate\_BD(new\_controllers)$         ▷ Use AE or RAED to generate BDs
8:         ADD ($collection, new\_controllers$)
9:         **if** $iteration \% training\_frequency == 0$ **then**
10:             $train\_network(collection)$
11:             $all\_controllers \leftarrow extract\_all(collection)$
12:             $generate\_BD(all\_controllers)$
13:             ADD ($collection, all\_controllers$)
14:         $L \leftarrow update\_distance\_threshold(L, collection, collection\_target\_size)$
15:
16: **procedure** ADD($collection, controllers$)         ▷ Attempt to add controllers to collection
17:     **for** controller in controllers **do**
18:         $distance \leftarrow get\_BD\_distance\_to\_nearest\_controller(controller, collection)$
19:         **if** $distance > L$ **then**            ▷ Add controller if sufficiently novel
20:             $add\_to\_collection(controller)$

---

| Parameter | Value |
|---|---:|
| Behavioural Descriptor Dimensionality | 2 |
| Mutation Rate | 0.1 |
| Cross-Over Rate | 0.1 |
| Mutation Operator | Polynomial (Dobnikar et al., 1999) |
| Cross-Over Operator | SBX (Deb & Agrawal, 1994) |
| $\eta_m$ (Mutation Parameter) | 15 |
| $\eta_c$ (Cross-Over Parameter) | 15 |
| Collection Target Size | 8000 |
| AURORA controller Range of Possible Values | [0, 1] |
| Behavioural Descriptor Size | 2 |
| Number of Newly Generated controllers each Iteration | 256 |
| First AE / RAED Training iteration | 10th Iteration |
| Frequency of AE / RAED Training | Every 10 Iterations |
| Number of Iterations (Object Arrangement Task) | 4000 |
| Number of Iterations (Air-Hockey Task) | 15000 |

Table 1: Hyperparameters for the AURORA implementation.

## B.2 AE AND RAED HYPERPARAMETERS

In the object arrangement scenario, the AE is made up of 4 convolutional layers including the output layer in both the Encoder and Decoder network. RAED's Encoder consists of 3 fully connected layers. The Decoder in RAED consists of 6 transposed convolutional layers. The network architecture details are given in Tab. 2.

| Layer | Number of Filters / Neurons | Kernel Size | Stride |
|---|---|---|---|
| AE Encoder #1 | 2 | 9 x 9 | 1 |
| AE Encoder #2 | 4 | 8 x 8 | 2 |
| AE Encoder #3 | 6 | 7 x 7 | 1 |
| AE Encoder Output Layer | 2 | 7 x 7 | 1 |
| AE Decoder #1 | 6 | 7 x 7 | 1 |
| AE Decoder #2 | 4 | 7 x 7 | 1 |
| AE Decoder #3 | 2 | 8 x 8 | 2 |
| AE Decoder Output Layer | 1 | 9 x 9 | 1 |
| RAED Encoder #1 | 10 | / | / |
| RAED Encoder #2 | 20 | / | / |
| RAED Encoder Output Layer | 2 | / | / |
| RAED Decoder #1 | 10 | 3 x 3 | 1 |
| RAED Decoder #2 | 10 | 3 x 3 | 1 |
| RAED Decoder #3 | 20 | 4 x 4 | 2 |
| RAED Decoder #4 | 20 | 5 x 5 | 1 |
| RAED Decoder #5 | 40 | 5 x 5 | 2 |
| RAED Decoder Output Layer | 1 | 6 x 6 | 1 |

Table 2: AE and RAED architecture details.

In the air-hockey scenario, the AE Encoder consists of 6 convolutional layers The AE Decoder features 6 transposed convolutional layers. RAED's Encoder is made up of 7 fully connected layers and its Decoder contains 9 transposed convolutional layers. The architecture details used in the air-hockey task are given in Tab. 3.

| Layer | Number of Filters / Neurons | Kernel Size | Stride |
|---|---|---|---|
| AE Encoder #1 | 10 | 6 x 6 | 1 |
| AE Encoder #2 | 10 | 5 x 5 | 2 |
| AE Encoder #3 | 20 | 5 x 5 | 1 |
| AE Encoder #4 | 20 | 4 x 4 | 2 |
| AE Encoder #5 | 30 | 3 x 3 | 1 |
| AE Encoder Output Layer | 2 | 3 x 3 | 1 |
| RAED Encoder #1 | 20 | / | / |
| RAED Encoder #2 | 30 | / | / |
| RAED Encoder #3 | 40 | / | / |
| RAED Encoder #4 | 40 | / | / |
| RAED Encoder #5 | 30 | / | / |
| RAED Encoder #6 | 20 | / | / |
| RAED Encoder Output Layer | 2 | / | / |
| RAED Decoder #1 | 10 | 2 x 2 | 1 |
| RAED Decoder #2 | 10 | 3 x 3 | 1 |
| RAED Decoder #3 | 20 | 3 x 3 | 1 |
| RAED Decoder #4 | 20 | 3 x 3 | 1 |
| RAED Decoder #5 | 30 | 3 x 3 | 1 |
| RAED Decoder #6 | 40 | 3 x 3 | 1 |
| RAED Decoder #7 | 60 | 5 x 5 | 1 |
| RAED Decoder #8 | 60 | 5 x 5 | 2 |
| RAED Decoder Output Layer | 1 | 6 x 6 | 1 |

Table 3: AE-SMALL and RAED-LARGE architecture details.

Both networks are trained with the same training parameters given in Tab. 4. The networks are implemented using the PyTorch library[1].

| Parameter | Value / Definition |
|---|---|
| Batch Size | 256 |
| Learning Rate | 0.0001 |
| Optimisation Algorithm | Adam (Kingma & Ba, 2015) |
| Adam Betas | (0.9, 0.999) |
| Validation Set Size (% of total dataset) | 80% |
| Maximum Number of Training Epochs | 5000 |
| Early Stopping Criterion | Validation error larger than running mean of validation errors over 5 epochs AND most recent training loss smaller than at start of training |

Table 4: Network training details.

## C  SIMULATION DETAILS

Our simulated environments utilise the Box2D[2] physics simulation engine to model a robot arm mounted in the center of a table. The arm is controlled with servos setting the motor speed at each joint to reduce the angle error. We specify the Box2D specific simulation parameters in Tab. 5. The Box2D source files were modified to change a *#define* macro, to allow for simulated pucks to bounce instead of gliding along any walls after impact. The change was made in file *box2d/include/box2d/b2_settings.h* to update the value of *#define b2_velocityThreshold* from *1.0f* to *0.0f*.

| Configuration Element | Dimensions (half sizes per Box2D convention) / Value |
|---|---|
| Gravity | (0, 0) |
| Robot Base | 0.0375 x 0.0375 |
| Arm Segment | 0.1875 x 0.015 |
| Arm Segment #1 Density | 1 |
| Arm Segment #2 Density | 0.666666 |
| Arm Segment #3 Density | 0.444444 |
| Arm Segment #4 Density | 0.296296 |
| Arm Friction (All components) | 0.8 |
| Servo Maximum Motor Torque | 1 |
| Table Dimensions | 5 x 5 |
| Table Wall Friction | 0.8 |
| Simulation Duration | 10 seconds |
| Environment Object Radius | 0.25 |
| Environment Object Friction | 0.8 |
| Environment Object Restitution | 0.8 |
| Scenario 1 Puck Density | 0.2 |
| Scenario 1 Maximum Random Force | 5 |
| Scenario 1 Non-Random Puck Start Position | (3.55, 3) |
| Scenario 2 Object Density | 0.8 |
| Scenario 2 Maximum Force | 70 |
| Scenario 2 Linear Damping | 1.8 |
| Scenario 2 Object 1 Start Position | (3.55, 3) |
| Scenario 2 Object 2 Start Position | (2.55, 1.7) |

Table 5: Box2D configuration parameters.

---

[1] https://pytorch.org/cppdocs/frontend.html
[2] https://box2d.org/documentation/index.html

