# OpenReview forum: "On the Importance of Distraction-Robust Representations for Robot Learning"
_ICLR.cc/2021/Conference — Reject_

### Official Review · AnonReviewer4 · 2020-10-26
**Unclear take-home message**

**Rating:** 4
**Confidence:** 4

**Review:**

#### Summary

Distractions are alterations of states or observations outside the control of the agent.  Conventional learning methods tend to be quite sensitive to them.  This work proposes a method for learning representations robust to such distractions using an encoder-decoder architecture whose encoder uses actions as input, instead of an observation as conventional autoencoders do.  In addition, the paper evaluates two variants of this method that employ an additional regularization term, one being the KL divergence as used by variational autoencoders, the other the $L_2$ norm of the latent activation vector (to avoid penalizing its variance as the KL divergence does).

The method is evaluted as the generator of the latent space ("Behavioral Descriptors") of an AURORA (Cully 2019) setup where randomly-evolved controllers are accepted or rejected based on their diversity in latent space.  Two example tasks using a planar, 4-DoF robot arm and 40$\times$40 image observations are used. The first task involves pushing two high-friction objects; distractions are externally-imposed displacements of one object.  The other task involves hitting one low-friction object (air-hockey style); distractions consist in the insertion of a second, random puck trajectory into the observation.  By most metrics, the method is shown to be effective at increasing robustness of the latent representation to distractors.

#### Strengths and Weaknesses

Strengths include:

* The stated problem is important and arises in many contexts.
* The key part of the method, learning representations by encoding actions as opposed to observations, makes a lot of sense in this context.
* The proposed solution can be plugged into existing representation-learning methods, as is done in the evaluation.
* For the most part, the paper is well organized, well written, and clear.
* The appendix provides implementation details.

Weaknesses include:

* As described, the method can only be used to work with controllers that operate on a state space distinct from the learned latent space.  Thus, the latent space can only used for purposes other than control, such as selecting controllers based on diversity criteria.  In practice, one usually wants to use distraction-robust latent representations as input to (distraction-robust) controllers.  This work does not appear to be applicable to this use case.  Thus, the practical scope and usefulness of this method appear to be rather limited, and the ultimate value of the learned representation is unclear.  In this light, the title and abstract can easily be misread as promising more than the article delivers ("We propose an Encoder-Decoder architecture that produces representations that allow the learning outcomes of robot control tasks to remain unaffected by these distractions.").
* In Figure 6b, the RAED methods perform rather poorly in absolute terms.  No explanation is given (besides an explanation of why the AE method performs relatively well here).
* The evaluation considers two quite similar tasks, and the results are not fully conclusive.  Thus, it is hard to extract a clear take-home message from this paper.
* As formulated, the input vector to the encoder represents an action sequence encoding an entire episode.  In the experiments shown this did not pose a problem as an episode consisted of a single action.  However, it is unclear how the method would perform on multi-action tasks.  Encoding entire episodes would not scale and is not possible on non-episodic tasks.  It is unclear how the method would be used and how it would perform if applied on single actions in multi-action tasks.
* The technical contribution is limited, quite close to the smallest publishable unit (if such is quantifiable), especially considering the inconclusiveness of the experimental results.

#### Justification of Recommendation

The technical contribution is minor, its applicability is quite limited, and the results are not fully conclusive.  This weakens the take-home message of the paper.

#### Questions for the Rebuttal

* Please comment on the weaknesses listed above.
* In Sec. 4.1, observations are taken at the *end* of a simulation, and DIs are injected at the *start* of a simulation.  Does this mean that the disturbed state is never directly observed but manifests itself only indirectly at the next simulation? That is, the state persists across simulations and is not (randomly) initialized for each simulation?
* Figures 3 and 6 show interquartile ranges. Since these are computed over only 5 repetitions, I take this to mean that the ranges shown exclude the minimum and the maximum value everywhere.
* In Fig. 6b, *failing to move both objects* is apparently intended to mean *failing to move one of the two objects*.  The latter formulation is more clear, as the former logically includes failing to move any object.
* Why was RAED-KLPENALTY excluded from evaluation on the air-hockey task? I don't think the reason given justifies making the results appear incomplete.

---

### Official Review · AnonReviewer3 · 2020-10-27
**Official Blind Review**

**Rating:** 4
**Confidence:** 4

**Review:**

This paper introduces the concept of Distraction-Robust representation learning in their words. Their algorithm embeds autoencoder (which weirdly takes action as input. to restore images) into the AURORA framework(a kind of genetic algorithm). They conduct two toy experiments to show the effectiveness of their algorithm.


Strenght:
- The paper is well-written and easy-to-follow.
- The paper is well-motivated. Learning a distraction-robust representation is important for robot vision tasks, e.g. object tracking.

Weakness:
- This work lacks novelty. The core algorithm is just a regular autoencoder that has been well studied, see [1] for a thorough clarification. And the framework of the genetic algorithm mainly draws on AURORA.
- The toy experiments are not convincing. The setting is relatively clean/simple, comparing to most real-world scenarios, in the aspect of system dynamics, appearance, and etc. For the robot learning field,  the tasks should be defined closely to real-world robot setting such as object manipulation[2], object tracking[3], etc, either in a simulation environment or the real world.  In my opinion, it is necessary to conduct an experiment to demonstrate the generalization of the proposed method. For example, the simplest idea is training/testing the model in environments with random nature noise (placing different nature images/videos as the background).

Ref:

[1] Kingma D P, Welling M. An introduction to variational autoencoders[J]. arXiv preprint arXiv:1906.02691, 2019.

[2] Wilson, Matthew, and Tucker Hermans. "Learning to manipulate object collections using grounded state representations." Conference on Robot Learning. PMLR, 2020.

[3] Luo, Wenhan, et al. "End-to-end active object tracking and its real-world deployment via reinforcement learning." IEEE transactions on pattern analysis and machine intelligence 42.6 (2019): 1317-1332.

---

### Official Review · AnonReviewer1 · 2020-10-28

**Rating:** 3
**Confidence:** 3

**Review:**

This paper approaches the problem of representation learning for robotic learning, with a focus on being robust to potentially misleading distractor objects. Concretely, they propose a method which learns a representation of sequences of actions by reconstructing a future state, and uses this representation of actions to learn skills. They show that this leads to some improvement in identifying more diverse "skills" in settings where there are external distractors.

Pros:
+ The problem of learning representations for robotic learning which are robust to distractors is very important and relevant.
+ The proposed approach is simple to implement.

Cons:
I have several concerns about the paper, concretely regarding (1) the clarity of presentation, (2) the correctness/generality of the proposed method, and (3) the conclusions drawn from the experimental results.

(1): There are serious issues with the clarity of the paper.

First, much of the abstract/introduction motivates the challenge of learning representations of visual inputs that are robust to distractors. The related work also covers some prior works on this topic in general the paper seems to be proposing a technique to address this challenge. But upon reaching the method, it appears that the paper is not concerned with representation of states at all, but rather representations of sequences of actions with no state information. This a completely disconnected from the motivation/problems presented in the introduction/related work. It is not clear what the objective of the paper is - is it to learn efficient state representations for reinforcement learning? is it to learn a generative model over actions? What is the goal of such a representation, better reinforcement learning? These questions should be clearly answered early in the paper.

Second, this representation of action sequences is not actually used for any specific RL or task learning. Instead it is used with a method for unsupervised skill discovery AURORA, which tries to explore + build a set of controllers. If the motivating problem is representations robust to distractors, why use a unsupervised skill discovery framework like this? It seems like the problem of robust representations can be studied in a much simpler way (evaluating some task performance in the face of distractors). On the other hand, if the goal is a better exploration + skill discovery framework using a compact action representation, that should be laid out from the beginning of the paper.

(2): I have some doubts about the correctness of the method (or the implicit assumptions made).

First, The idea of the paper is to encode a sequence of actions into a latent distribution from which the future observation can be recovered. Is this not simply learning a forward dynamics model, but under the very restrictive assumption that the future state does not depend on the current state? Is this work considering an MDP? The necessary preliminaries are not stated, so its not clear what problem setting the paper considering.

Second, the way the KL penalty is used on the latent of action sequences does not seem correct. The encoder is outputting a latent distribution over action sequences. The reason to have this be a distribution is to capture stochasticity in the learned representation. But the authors then argue they don't want the latent distribution to have high variance, so they only use a penalty on the mean. In that case, why have the encoder output a distribution?

(3): Experimental conclusions.

The only experimental comparison is the an auto encoder on state. Since the proposed method is not encoding state, it seems obvious that the representation would be robust to state perturbations, while the auto-encoder on the state would not be. In fact its not clear why the auto-encoder on state is an effective baseline, if the goal is to learn sets of controllers composed of action sequences. Wouldn't simply learning an auto-encoder on action sequences be a more effective baseline?

Also more of a minor point, but the points to not considering realistic/stochastic settings with distractors as a limitation of prior work, but the domains considered in this paper are far from realistic.

---

### Official Review · AnonReviewer2 · 2020-10-28
**Confusing and possibly flawed method, difficult to parse**

**Rating:** 3
**Confidence:** 4

**Review:**

Summary:
This paper seeks to learn a representation space for trajectories that discards state elements that are not affected by the agent. To do so, it proposes an objective to predict the last observation in a trajectory from the sequence of actions taken in the trajectory.

Pros:
Tackles the problem of generating diverse task-relevant behaviors
Cons:
The motivation and the presentation of the method are very confusing
I am not convinced that the objective is a reasonable one (see comments below)
Experiments are very toy, and the analysis is hard to parse

Detailed Comments
Introduction: “When the dimensionality of the representation is smaller than that of the input data, some information is lost when creating the representations.” Dimensionality reduction does not necessarily imply information reduction. Also, the goal of representation learning need not be dimensionality reduction (sparse coding, for example).

I do not think that terms such as “representation learning” and “reinforcement learning” should be capitalized. They are not proper nouns.

It’s strange that “a” refers to a sequence of actions, but “x” refers to a single observation.

It seems like in Equation (2) you would want to predict a distribution over final states, to account for stochastic dynamics. Also, it seems like there is an implicit assumption that all objects need to start in the same place. For example, say you want a robot to pick up a cup. Depending on the starting position of the cup, the same sequence of actions could lead to very different final states.

The discussion in Section 3 about choosing the form of the penalty P is quite confusing and not rigorous.

It’s not clear until the experiments section of the paper that the goal is to produce diverse trajectories. This makes the beginning of the paper very hard to understand.

How is the representation learned by the auto-encoder of each observation transformed into a behavioural descriptor for the whole trajectory? Is the behavioural descriptor simply taken to be the embedding of the final observation? If this is the case, then how do the DIs affect the auto-encoder baseline at all?

The air-hockey plot seems to be missing a line?

Recommendation:
Reject
The method as presented seems flawed, and the experiments are toy and hard to make sense of. The paper overall is very difficult to understand.

---

### Author Response · Authors · 2020-11-24
**Thank you**

Dear reviewers,

Thank you for your valuable comments. We are glad that you generally recognise the importance of the problem of learning representation in the presence of distractors.
We are taking good note of your comments and we will continue to work on the paper to make it stronger.

---

### Decision · Program_Chairs · 2021-01-07
**Final Decision**

**Decision:**

Reject

**Comment:**

While the general idea of the paper is certainly interesting and highly relevant, there is consensus that the paper cannot be published in the current form.
There were serious concerns about
- the correctness and generality of the method
- clarity of presentation
- experimental evaluation

The authors graciously accepted the feedback, we wish them all the best in thoroughly revising and resubmitting the paper.